# Epibatidine: A Promising Natural Alkaloid in Health

**DOI:** 10.3390/biom9010006

**Published:** 2018-12-23

**Authors:** Bahare Salehi, Simona Sestito, Simona Rapposelli, Gregorio Peron, Daniela Calina, Mehdi Sharifi-Rad, Farukh Sharopov, Natália Martins, Javad Sharifi-Rad

**Affiliations:** 1Student Research Committee, School of Medicine, Bam University of Medical Sciences, Bam 44340847, Iran; bahar.salehi007@gmail.com; 2Department of Pharmacy, University of Pisa, Via Bonanno 6, 56126 Pisa, Italy; simona.sestito@for.unipi.it (S.S.); simona.rapposelli@unipi.it (S.R.); 3Interdepartmental Research Centre for Biology and Pathology of Aging, University of Pisa, 55126 Pisa, Italy; 4Department of Pharmaceutical and Pharmacological Sciences, University of Padova, Via Francesco Marzolo, 5, 35131 Padova (PD), Italy; gregorio.peron@ub.edu; 5Department of Clinical Pharmacy, University of Medicine and Pharmacy Craiova, Craiova 200349, Romania; calinadaniela@gmail.com; 6Department of Medical Parasitology, Zabol University of Medical Sciences, Zabol 61663-335, Iran; 7Department of Pharmaceutical Technology, Avicenna Tajik State Medical University, Rudaki 139, Dushanbe 734003, Tajikistan; shfarukh@mail.ru; 8Faculty of Medicine, University of Porto, Alameda Prof. Hernâni Monteiro, 4200-319 Porto, Portugal; 9Institute for Research and Innovation in Health (i3S), University of Porto, 4200-135 Porto, Portugal; 10Zabol Medicinal Plants Research Center, Zabol University of Medical Sciences, Zabol 61615-585, Iran; 11Department of Chemistry, Richardson College for the Environmental Science Complex, The University of Winnipeg, Winnipeg, MB R3B 2G3, Canada

**Keywords:** epibatidine, nicotinic acetylcholine receptors, analgesics, ABT-594, ABT-418

## Abstract

Epibatidine is a natural alkaloid that acts at nicotinic acetylcholine receptors (nAChRs). The present review aims to carefully discuss the affinity of epibatidine and its synthetic derivatives, analogues to nAChRs for α4β2 subtype, pharmacokinetic parameters, and its role in health. Published literature shows a low affinity and lack of binding of epibatidine and its synthetic analogues to plasma proteins, indicating their availability for metabolism. Because of its high toxicity, the therapeutic use of epibatidine is hampered. However, new synthetic analogs endowed from this molecule have been developed, with a better therapeutic window and improved selectivity. All these aspects are also discussed here. On the other hand, many reports are devoted to structure–activity relationships to obtain optically active epibatidine and its analogues, and to access its pharmacological effects. Although pharmacological results are obtained from experimental studies and only a few clinical trials, new perspectives are open for the discovery of new drug therapies.

## 1. Introduction

Epibatidine (exo-2-(6-chloro-3-pyridyl)-7-azabicyclo-[2.2.1]heptane) is a toxic alkaloid isolated and identified from *Epipedobates tricolor* skin, an Ecuadorian poison frog used by indigenous tribes in darts for hunting [1]. Although it was discovered in 1974 by Daly and Myers, epibatidine formula C_11_H_13_N_2_Cl and its chemical structure was only established in 1992 using nuclear magnetic resonance spectroscopy [2].

This alkaloid is widely considered an analgesic agent, with a potency ranging from 100- to 200-fold higher than the analgesic opioid morphine and 30 times higher than nicotine. The analgesic mode of action of epibatidine is attributed to its interaction with nicotinic acetylcholine receptors (nAChRs), and not with the opioid receptors. Nicotinic acetylcholine receptors are transmembrane oligomeric ligand-gated ion channels and are expressed both central and peripherally. The muscle nAChRs is a pentamer and it consist of a combination of four classes subunits (subunit α expressed in two copies and the other three subunits β, γ, δ as single copies) that form a transmembrane aqueous pore. The neuronal nAChRs are pentameric structures with a combination of only two classes of subunits (α and β). The most abundant heteromeric nAChRs subtypes in the brain are composed of two αand three β subunits (i.e., (α4)_2_(β2)_3_) [3]. Particularly, in mammalian central nervous system (CNS) the heterodimeric nAChR subtypes α4β2, and the monomeric subtypes α7 are the most expressed. Epibatidine has high affinity to nAChRs for the α4β2 subtype, being a potent, but non-selective (α4β2 *K*_i_ = 40 pM; α7 *K*_i_ = 20 nM) nAChR agonist. The inhibitory constant (*K*_i_) of a drug is known to cause the inhibition of a cytochrome P450 enzyme and have to do with the concentration needed to reduce the activity of that enzyme by half. The *K*_i_ is reflective of the binding affinity for a drug [4]. 

Unfortunately, its broad-spectrum of activity induces several off-targets effects in several districts, such as in CNS as well as in respiratory, gastrointestinal, and cardiovascular functions, precluding any therapeutic development [5]. Thus, epibatidine toxicity could be related to its ability to activate not only the central neuronal α2β2, but also the ganglionic α3β4 nAchR.

The great interest in epibatidine arises with the discovery of the analgesic activity mediated by non-opioid receptors. This mode of action is a good alternative to induce analgesia with no risk of dependence, tolerance and psychological dependence. The lack of opioid-receptors mediated activity has been proved by the co-administration of naloxone, an opioid antagonist. The experiment performed by Qian et al. [6] proved that the epibatidine-induced anti-nociception in mice and rats, is not influenced by naloxone, but the effective therapeutic dose is close to the lethal dose and, thus, the therapeutic safety interval is narrow. The main clinical signs of epibatidine toxicity includes activation of exocrine secretions (rhinorrhea, sialorrhea, lacrimation), seizures, hypertension, and muscle paralysis. These effects occur because epibatidine binds to nAChRs in the CNS, as well as to nAChRs in skeletal neuro-muscular junctions [7]. For instance, Thompson et al. [8] demonstrated that epibatidine binds to α-7 nicotinic receptors (α7nACh) present both peripherally and centrally in the nervous system. These central receptors are involved in various neurological conditions, such as schizophrenia, Parkinson’s, and Alzheimer’s disease (AD), but also in other important physiological functions, including neuroprotection, memory and learning, and pain control [9]. For this reason, new ligands of these receptors have been investigated as upcoming pharmacological research tools or as possible drugs.

In light of these aspects, it has been found that the new neuronal nAChRs agonists synthesized have a greater affinity to the α4β2 receptor subtype of the CNS and low affinity to neuromuscular junction nAChRs. Thus, Abbott Laboratories synthesized a neuronal nACh receptor agonist named ABT-594 (tebanicline or ebanicline) derived from the epibatidine structure, which binds preferentially to α4β2 receptor subtype [10]. More recently, other studies demonstrated other pharmacological effects of epibatidine. For instance, Shimizu et al. [11] reported increases in adrenaline and noradrenaline concentrations after intracerebroventricular injection of medium doses of epibatidine in rats. Moreover, it has been shown that nicotinic receptors α4β2 activation in brain produce elevated secretion of catecholamines from adrenal glands [11]. Unexpected outcomes were obtained by Green et al. [12] in recent research. They demonstrated, in an *in vivo* study using a pregnant goats model, that epibatidine dosed at 0.002 mg/kg did not completely inhibit fetal movements, but produced intoxication with clinical signs (salivation, muscle fasciculation, urination). These results are important because the complete inhibition of fetal movements is associated with an increased risk of teratogenicity [12].

## 2. Pharmacokinetics of Epibatidine and Its Synthetic Derivatives

Epibatidine biological targets and mechanisms of action are now well established [13], but little is known on its pharmacokinetics. Although some parameters, like intestinal absorption, distribution through the blood–brain barrier (BBB) and metabolism by the cytochrome CYP450 enzyme family, can be predicted by means of informatic tools (http://lmmd.ecust.edu.cn/admetsar1) (data about epibatidine can be found on DrugBank (DB07720 at www.drugback.ca)), experimental *in vitro* and *in vivo* data still remain scarce. London et al. [14] reported that epibatidine reaches the highest distribution in thalamus and upper colliculus in the brain 30 min after tail-vein administration in rats. A slow clearance was also observed, with epibatidine still present in brain 4 h post administration [14]. More recently, Javors et al. [15] investigated epibatidine absorption and distribution using a mouse model. A dose of 0.1 mg/kg epibatidine was administered intraperitoneally to C57BL/10J mice and the alkaloid concentration was monitored in plasma samples collected 10 min post-injection [15]. Significantly different amounts of epibatidine were observed between male and female mice, with concentrations of 7.3 ng/mL for males and 37.1 ng/mL for females [15]. However, this discrepancy was not discussed further. Similar results were then obtained by Shiraishi and coauthors [16] using a rat model, where a plasma concentration of 5.8 ng/mL was detected following intraperitoneal injection of 10 µg/kg epibatidine [16].

Data on epibatidine metabolism are scarce. Preliminary observations were published in 2000 by Watt et al. [17]. This research investigated *in vitro* the route of metabolism of epibatidine (+) and (−) enantiomers. The results showed the formation of the diastereoisomeric N-oxides for (+) for (+) epibatidine and hydroxylation of the azabicycle for (−) epibatidine [17].

Studies on epibatidine metabolism and metabolites excretion have undergone gradual decreased in number due to its severe side effects. More recently, efforts have been made to address the pharmacokinetics of new derivatives of epibatidine with reduced toxicity, as reported by Heugebaert et al. [18]. The authors presented five epibatidine analogs containing a substituent on the azabicyclo[2.2.1]heptane bridgehead, namely a ketone linker, an OH linker, an aminomethyl linker, and two ethyl linkers containing an OH and a NH_2_ group, respectively. Only ketone-binding epibatidine maintained chlorine in the aromatic ring. However, the authors focused their attention only on an aminomethyl- and two ethyl linkers containing an OH and a NH_2_ group, due to their higher affinity for α4β2 nAChR [18], and for them studied the *in vitro* pharmacokinetics. Both compounds showed a low binding affinity to plasma proteins, meaning that they are available for metabolism [18]. Using isolated rat hepatocytes, an evaluation of metabolic stability was carried out and the results indicated compounds with an OH linker as metabolically stable, while the metabolism of an aminomethyl linker seemed to be concentration-dependent and faster than that of control (verapamil), thus hampering its t_1/2_ (<2 min, compared to t_1/2_ = 175 min for OH binding compound; the half-time (t_1/2_) is a pharmacokinetic parameter, being defined as the required time to decrease half of the concentration of an active substance in plasma) [18].

Other epibatidine derivatives have been synthetized over the past few years and evaluated for a possible use as radioligands for neuroimaging by positron emission tomography (PET) of α4β2 nAChR. Among others, radiolabeled derivatives of flubatine (Figure 1), (+)- and (−)-[^18^F] enantiomers of flubatine seem to represent the most suitable candidates and have been examined in several studies to investigate their metabolic fate both *in vitro* and *in vivo*. In a study by Patt et al. [19], plasma protein binding, metabolism, and activity distribution between plasma and whole blood of (−)- enantiomer of [^18^F]flubatine were investigated in 21 AD patients and 20 healthy controls. The obtained results showed that the amount of (−)- enantiomer of [^18^F]flubatine that bonded to plasma proteins was 15%, and was found to be poorly metabolized, with almost 90% of the administered dose remaining unchanged after 90 min post-injection. In addition, radioactivity distribution between plasma and whole blood only slightly changed over time, from 0.82 at 3 min post-injection to 0.87 at 270 min post-injection, indicating the contribution of only a small amount of metabolites [19]. In a more recent study by Ludwig and coauthors [20], the metabolic fate of (+)- enantiomer of [^18^F]flubatine was investigated in a clinical study involving patients with early AD. The same experiment was carried out for non-radiolabeled (+)- enantiomer of flubatine in pigs. For the latter, the six major metabolites detected in urine were formed by monohydroxylation at different sites of the azabicyclic ring system. An intermediate metabolite underwent glucuronidation, both *in vitro* and *in vivo*, and was detected in both plasma and urine [20]. In humans, it was observed that 30 min post-injection, 95.9% of the administered (+)- enantiomer of [^18^F]flubatine dose was present in plasma as unmodified, while 95.1% were present in urine, as further proof of the metabolism stability of these derivatives. As observed in pigs for (+)- enantiomer of flubatine, the main metabolites detected in plasma and urine were formed by monohydroxylation at the azabicyclic ring system (0.0−3.8% in plasma and 0.0−4.9% in urine, 30 min post-injection) and glucuronidation (0.0–4.0% in plasma and 0.4–10.7% in urine, 30 min post-injection) [20].

## 3. Role of Epibatidine in Health

Unfortunately, the broad spectrum of epibatidine activity on a nAChRs induces several off-targets effects in several districts, such as in CNS, respiratory, gastrointestinal, and cardiovascular functions, resulting in a limited therapeutic index (~4), thus, precluding any therapeutic development [5]. In particular, epibatidine toxicity may be related to its ability to activate not only central neuronal α2β2, but also ganglionic α3β4 nAchR. Therefore, the research community switched to modifying the epibatidine structure [21,22] to obtain analogues with a better pharmacological activity/toxicity ratio and selectivity for different nAchR subtypes (e.g., α2β2 compared to α3β4) [23,24].

In 1997, Badio et al. [25] synthetized (±)-epiboxidine, which contains a methylisoxazolyl ring replacing the chloropyridinyl ring of epibatidine (Figure 2). The rationale behind this chemical manipulation was inspired by ABT-418 (3-Methyl-5-[(2S)-1-methyl-2-pyrrolidinyl]isoxazole hydrochloride), a nicotine analogue with a methylisoxazolyl ring instead of a pyridine one. ABT-418 is a α4β2 selective full agonist, that has been used in a study for the treatment of cognitive dysfunction [26]. In that research, epiboxidine showed of about 10-fold less potency than epibatidine, but about 17-fold greater than ABT-418 in inhibiting [^3^H] nicotine binding to α4β2 nicotinic receptors on rat cerebral cortical membranes. Although the epiboxidineantinociceptive activity was about 10-fold less potent than epibatidine, isoxazolylbioisoster showed reduced toxicity in mice [25].

Later, compound ABT-594, [(R)-5-(2-azetidinylmethoxy)-2-chloropyridine], also known as tebanicline, was synthetized by Abbott [27] and was the first-in-class compound to reach clinical trials evaluation for neuropathic pain treatment (Figure 2). Preclinical studies in animals using ABT-594 showed no adverse effects of opioids: pharmacological dependence, respiratory depression and a greater analgesic effect. It was able to selectively inhibit the afferent transmission of acute (chemical or thermal), chronic or peripheral diabetic neuropathic pain signal [28,29]. The analgesic doses of ABT-594 ranged between 0.04 and 0.12 mg/kg, intraperitoneal ABT-594 administration at maximum doses produced adverse effects, such as hypothermia, hyperreactivity, and deficiency in voluntary movement control and body posture (ataxia) [30]. Preclinical evaluations in acute thermal pain (hot-plate) and persistent pain (formalin test) models indicated an efficient antinociceptive activity, predominately mediated by central neuronal nAChRs, although a contribution of peripheral nAChRs is not completely excluded [31].

Over the last two decades, the concept of multimodal analgesia has emerged to treat acute or chronic pain. It consists of an administration of two analgesic drugs from different pharmacological classes, with known and different mechanism of action, at smaller doses with the consequent reduction of adverse effects. This concept was used in a study carried out by Munro et al. [32], who associated ABT-594 with gabapentin, an antiepileptic drug which interacts with high affinity with voltage-sensitive Ca^2+^ channels in the brain [33], morphine (μ-opioid receptor agonist) and duloxetine (a potent inhibitor of neuronal reuptake of serotonin and norepinephrine) in rat formalin test. In the combination with gabapentin, mecamylamine administration (a non-selective nAChR antagonist) at the same time as ABT-594, removes the analgesic effect. Thus, supra-spinal action site at the raphe magus nucleus was then confirmed [34]. The results obtained by authors were surprising since the pharmacologically inactive doses of ABT-549 (0.01–0.3 mg/kg) associated with lower doses of each analgesic had proven synergistic analgesic effects [32].

At the outset, ABT-594 did not show sedative-like effects on electroencephalography typically induced by opioids, but additional studies have highlighted a partial involvement of opioid receptors through indirect activation [35].

The observed side effects (decrease in body temperature and impaired motor coordination) were significantly reduced with repeated administration [31]. A phase 2, randomized, multicenter, double-blind, placebo-controlled study with ABT-594 was conducted to assess the analgesic efficacy and safety profile in patients affected by diabetic peripheral neuropathic pain [29]. In all groups of patients treated with different doses (150, 225 and 300 µg), ABT-594 was able to induce a significant decrease in pain severity, in agreement with preclinical observations. Unfortunately, withdrawal rates due to adverse events were significantly higher in the three groups of ABT-594-treated patients, who presented nausea, dizziness, vomiting, abnormal dreams, and asthenia, consistent with the side effects profile of the class.

However, ABT-594 may be potentially useful as adjunctive therapy or in combination, since low doses did not activate α3-containing nAChRs, which are related with various autonomic nervous system side effects, such as cardiovascular and gastrointestinal [36].

Additional investigations on epibatidine and its analogues have revealed a high therapeutic potential in further diseases, as well as pain. In fact, as nAChRs are functionally involved in many essential CNS cellular mechanisms, including learning, anxiety, memory, and cognitive function, several nAchR ligands have been developed for the treatment of many pathologies, such as AD, attention deficit hyperactivity disorder (ADHD), anxiety, Parkinson’s disease, inflammatory bowel disorder, schizophrenia, depression, and many others [22]. For example, patients affected by neurodegenerative diseases, such as AD have a severe deficiency in nAChRs levels, suggesting a role for nAChR loss in cognitive decline in AD. This hypothesis is supported by PET studies in patients with AD, which demonstrated, on one hand, a correlation between a reduced nAChRs expression in brain and cognitive impairment grade and, on the other hand, an increase in nAChR density associated with better performance in cognitive tests after tacrine treatment [37]. Therefore, the availability of non-invasive tools for detecting the brain level of nAChRs could aid in early AD diagnosis, even at a pre-symptomatic stage, as well as in the development of personalized therapeutic regimens and in monitoring the efficacy of drug treatment [38].

Belluardo et al. [39] demonstrated that epibatidine induces fibroblast growth factor (FGF)-mRNA up-regulation, accompanied by an increase in FGF2 protein levels in rat brain, suggesting a neuroprotective effect of this natural alkaloid. In addition, epibatidine showed to be able to improve antioxidant and antiapoptotic effects, increasing heme oxygenase-1 (HO-1) expression, one of the main cytoprotective enzymes against oxidative stress [40].

Although no epibatidine derivative with neuroprotective activity has been synthetized so far, this option could represent a valid strategy to be followed. For instance, the clinical use of the nicotinic analogue ABT-418 has already been explored in moderate AD [41] and in less severe ADHD in adults [42]. The experimental *in vitro* and *in vivo* studies of epibatidine and its derivatives are summarized in Table 1.

In addition to its potential therapeutic role, epibatidine also represents an important research tool to investigate nAChR activity. Notably, [^3^H] epibatidine binds to nAchRs with very high affinity and extremely low non-specific binding. This radioligand easily crosses the blood brain barrier, binds reversibly to nAChRs and exhibits a moderately fast metabolism; therefore, it has been largely used for over 20 years as radioligand to study nAChRs [14,43]. From 1995, when [^3^H]epibatidine was firstly described, different radiolabeled (^11^C, ^18^F, ^76^Br and ^123^I) analogues were developed for *in vitro* and *in vivo* PET and single-photon emission computed tomography (SPECT) studies [37]. Recently, the exploration of the interaction of epibatidine with α7 receptor binding sites was carried out to lay a background for the design of specific epibatidine-based molecular probes, useful to investigate α7 function [8].

## 4. Conclusions

Epibatidine and its analogues have revealed a promising therapeutic potential in further diseases as well as pain. Data published show a low affinity and scarce binding of either epibatidine and its synthetic analogues to plasma proteins, indicating their availability for metabolism. However, quantitative data show that the amounts of both plasma and urinary metabolites are negligible compared to the amounts of underivatized compounds, indicating that, in general, they are not prone to metabolism. Due to severe gastrointestinal side effects, the first analogue of epibatidine, ABT-594, is not included in current pain therapies in humans. However, another new synthetic derivative of epibatidine ABT-418 is used in treatment of less severe ADHD in adult patients. ABT-418 has been well-tolerated by patients with minor side effects, such us nausea, dizziness, headaches, or skin irritations. Thus, epibatidine pharmacological effects open new perspectives in drug therapies and also represent an important research tool to investigate nAChR activity.

## Figures and Tables

**Figure 1 biomolecules-09-00006-f001:**
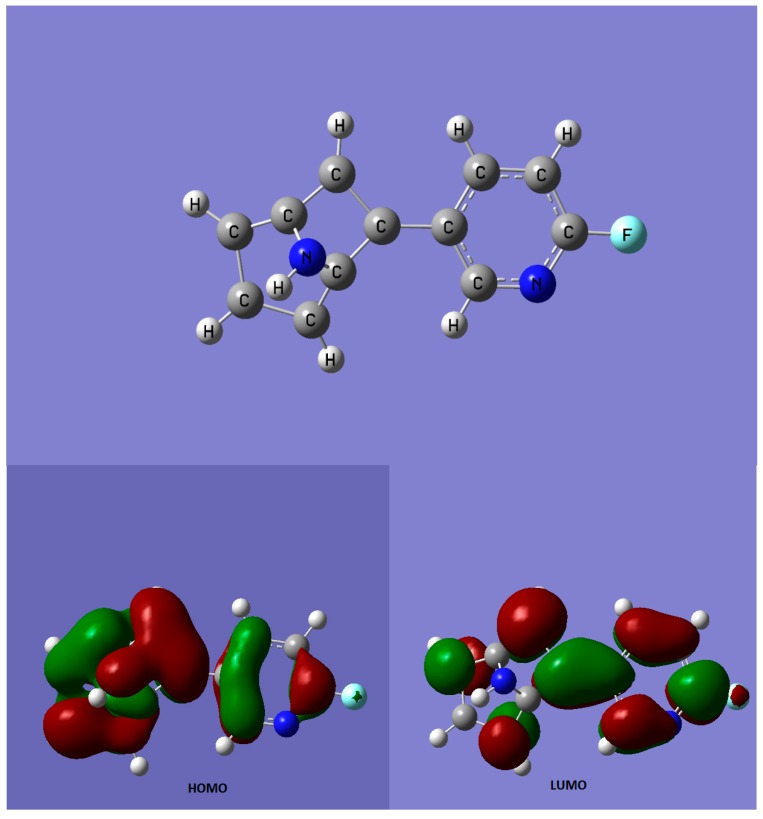
Chemical structure of flubatine (up) and charge distribution for the highest occupied molecular orbital (HOMO) and lowest unoccupied molecular orbital (LUMO) states (down). Methods: The ab initio calculations were performed with Gaussian 16w B.01 and modeled with GaussView 6.0.16 (Andreescu Labor Soft S.R.L., Bucharest, Romania). The optimization of the molecule and HOMO and LUMO states were calculated based on semi-empirical method PM6. It is easy to observe that the fluorine atom, for the HOMO statehas a positive charge; all the electrons are attracted on the rings. On LUMO state, the fluorine atom is partially shielded.

**Figure 2 biomolecules-09-00006-f002:**
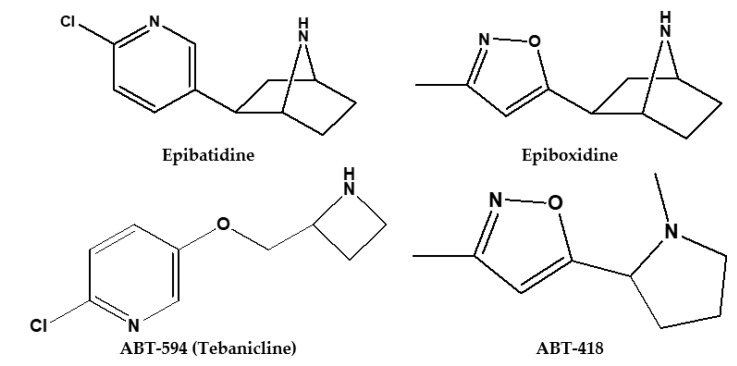
Epibatidine and its analogues.

**Table 1 biomolecules-09-00006-t001:** Pharmacological studies of epibatidine and its synthetic derivatives.

Compound	Type of Study		References
*In Vivo* Studies Using Animal Models			
Epibatidine	**Animal model**	**Primary outcomes**	
	Mice and rats	Antinociceptive	[6]
	Rats	Increased adrenaline and noradrenaline neuromediators	[12]
	Pregnant goats	Lack of completely inhibition of fetal movement	[11]
	Rats	Neuroprotective	[39]
ABT-594 tebanicline	Rodent pain models (rats, mice)	Antinociceptive	[27,28,30,31,35]
	Rat formalin test	Analgesic (multimodal analgesia)	[32]
Epiboxidine	Rats	Cognitive disfunction treatment	[26]
	Mice	Antinociceptive	
***In vitro* studies using cell lines**			
Epibatidine	Bovine chromaffin cells	Antioxidant, antiapoptotic	[40]
**Human clinical studies**			
ABT-594tebanicline	**Study design**	**Primary outcomes**	[29]
	Randomized, multicenter, double-blind, placebo-controlled study (phase 2)	Analgesic in diabetic patients with neuropathic pain	
ABT-418Epiboxidine	Double-blind, placebo-controlled study	Cognitive enhancement in moderate Alzheimer’s disease	[41]
	Double-blind, randomized, placebo-controlled, crossover trial	Increased attention in deficit hyperactivity disorder (ADHD) in adults	[42]

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
