# Peer review of "Epibatidine: A Promising Natural Alkaloid in Health"

_biomolecules, 2018, doi:10.3390/biom9010006_

Reviewer 1 Report

The manuscript by Salehi et al. is a fairly complete review of epibatidine.  In places it does need minor editing and more detail.  Tables summarizing some of the research findings would be useful.

 Comments

Line 26 delete renowned

Line 26 I would say …natural alkaloid that acts at nicotinic acetylcholine receptors.  Not sure what an analgesic attribute is.

Line 30 replace scare with “lack of”

Line 31 delete however start sentence with quantitative

Line 33 delete “in addition”

Line 36 only in rats is incorrect there have been studies of epibatidine conducted in other species.

Line 40 Introduction The authors refer to alpha 4 beta 2 nAChR frequently in this manuscript, they should consider discussing more of the biology of this receptor to provide the reader with some background.

Line 42 I’m fairly positive that all Ecuadorians are not using epibatidine to poison arrows for hunting.  Be more specific.

Line 43 probably should be Daly and Myers

Line 46 wordy use potent or potency only once

Line 48 I would say that epibatidine acts at nAChR and not at opioid receptors.  Not opioid receptors is plural because there is more than one type of opioid receptor.

Line 57 be more specific, nAChR are ligand gated ion channels.

Line 58 I would use nAChR instead of acetylcholine receptors.

Line 59 I would use nAChR instead of acetylcholine receptors.

Line 61 wordy delete therefore

Line 64  what is a morphine receptor, do you mean opioid receptor?

Line 65 naloxone is an opioid receptor antagonist, be specific

Line 69 epibatidine toxicity is cholinergic in nature the authors forgot salivation etc., be specific

Line 70 I like nAChR better than nicotinic receptors and most authors use that abbreviation in their writing.

Line 80 Sloppy pharmacology, α1β1γδ nAChR are fetal muscle type nAChR there are no analgesic effects.

Line 94 after the introduction of gabapentin the authors should mention where it acts at.  They do so with morphine and should so the same anytime they introduce a new drug. Do the same for duloxetine and mecamylamine.

Line 100 Epibatidine has been dosed to more than just rats.  For example, Green et al 2018 (Toxicon 144:61-67) dosed epibatidine to goats.

Line 135 The authors finally use nAChR.  Why not earlier in the manuscirpt!

Line 158 what is an AD patient?

Line 177 please revise “..on a nAChRs variety induces…”  not sure what is meant by that statement.

Line 196 The authors mention acute thermal pain and persistent pain models but did not define the models used.  Please define for the reader.

Line 199 first use of EEG without definition.

Line 234 What is FGF? 

Line 244 Why not use blood brain barrier instead of BBB. 

Line 247-248  PET and SPECT define these terms!

Line 248 -250 What is the spatial orientation of epibatidine binding?  Provide more information for the reader.

Author Response

Comments and Suggestions for Authors

The manuscript by Salehi et al. is a fairly complete review of epibatidine. In places it does need minor editing and more detail. Tables summarizing some of the research findings would be useful.

Answer: The most important pharmacological in vivoand in vitrostudies of epibatidine and its derivatives have been summarized in a table, Table 1, Section 3. 

Comments

Line 26 delete renowned

Answer: Done

Line 26 I would say …natural alkaloid that acts at nicotinic acetylcholine receptors. Not sure what an analgesic attribute is.

Answer: Done

Line 30 replace scare with “lack of”

Answer: Done

Line 31 delete however start sentence with quantitative

Answer: Done

Line 33 delete “in addition”

Answer: Done

Line 36 only in rats is incorrect there have been studies of epibatidine conducted in other species.

Answer: We have added goats too, thanks

Line 40 Introduction The authors refer to alpha 4 beta 2 nAChR frequently in this manuscript, they should consider discussing more of the biology of this receptor to provide the reader with some background.

Answer: Thank you. We added a paragraph in which the role of nAChRs subtype has been discussed

Line 42 I’m fairly positive that all Ecuadorians are not using epibatidine to poison arrows for hunting.  Be more specific.

Answer: Sorry. We have correctly cited the use of this poison from indigenous tribes. Thanks

Line 43 probably should be Daly and Myers

Answer: Yes, thank we corrected the citation

Line 46 wordy use potent or potency only once

Answer: done

Line 48 I would say that epibatidine acts at nAChR and not at opioid receptors.  Not opioid receptors is plural because there is more than one type of opioid receptor.

Answer: Thanks. Done

Line 57 be more specific, nAChR are ligand gated ion channels.

Answer: done

Line 58 I would use nAChR instead of acetylcholine receptors.

Answer: done

Line 59 I would use nAChR instead of acetylcholine receptors.

Answer: done

Line 61 wordy delete therefore

Answer: done

Line 64  what is a morphine receptor, do you mean opioid receptor?

Answer: Yes, thanks we have changed it with opioid receptor

Line 65 naloxone is an opioid receptor antagonist, be specific

Answer: done

Line 69 epibatidine toxicity is cholinergic in nature the authors forgot salivation etc., be specific

Answer: Thanks. We added other side-effects of epibatidine due also to an activation of exocrine secretions

Line 70 I like nAChR better than nicotinic receptors and most authors use that abbreviation in their writing.

Answer: done

Line 80 Sloppy pharmacology, α1β1γδ nAChR are fetal muscle type nAChR there are no analgesic effects.

Answer: Sorry for our carelessness, we have deleted this sentence

Line 94 after the introduction of gabapentin the authors should mention where it acts at.  They do so with morphine and should so the same anytime they introduce a new drug. Do the same for duloxetine and mecamylamine.

Answer: Thanks. Done

Line 100 Epibatidine has been dosed to more than just rats.  For example, Green et al 2018 (Toxicon 144:61-67) dosed epibatidine to goats.

Answer: Thank you we added the reference

Line 135 The authors finally use nAChR.  Why not earlier in the manuscirpt!

Answer: We changed nicotinic receptor with nAChRs along the entire manuscript. Thank you

Line 158 what is an AD patient?

Answer: We specified Alzheimer’s disease patient, thanks

Line 177 please revise “..on a nAChRs variety induces…”  not sure what is meant by that statement.

Answer: We delete variety

Line 196 The authors mention acute thermal pain and persistent pain models but did not define the models used.  Please define for the reader.

Answer: Acute thermal pain is hot-plate model, while persistent pain model is the formalin test assay. We specified also in the text.

Line 199 first use of EEG without definition.

Answer: Defined in the text, thank you

Line 234 What is FGF?

Answer: FGF is fibroblast growth factor. Wedefined the acronym in the text. Thanks

Line 244 Why not use blood brain barrier instead of BBB. 

Answer: done

Line 247-248 PET and SPECT define these terms!

Answer: done

Line 248 -250 What is the spatial orientation of epibatidine binding? Provide more information for the reader.

Answer: We discussed better the spatial orientation of epibatidine. This is the interaction mode of the molecule with the receptor that could be of help to identify new pharmacophore group needed to increase the selectivity.

Reviewer 2 Report

This manuscript reviews the biology of epibatidine, a natural alkaloid, and its synthetic derivatives. Epibatidine has analgesic effects through α4β2 nicotinic acetylcholine receptors, and could have potential as a non-addictive alternative to opioid analgesic agents. However, its use as an analgesic is limited by toxicity mediated through CNS and skeletal neuro-muscular junction α7 nicotinic receptors, which can cause seizures and muscle paralysis. The resultant narrow therapeutic window could be widened by the development of derivatives with improved selectivity towards the α4β2 acetylcholine receptor.

The manuscript contains a considerable amount of information about epibatidine and its derivatives, but the value is reduced by a lack of organization and by redundancy. The impression given is that a number of authors have contributed sections but no-one has taken the time to synthesize the information adequately into a coherent document. In addition, given the toxicity of epibatidine, it would seem appropriate to focus more on the derivatives and less on the parent compound.

Abstract: The abstract should include the limitations of epibatidine due to toxicity and the potential of synthetic analogs as analgesic agents through improved selectivity. The metabolism of epibatidine is of minor interest and could be omitted from the abstract.

Introduction: The introduction lacks focus. e.g. the introduction of synthetic strategies in Line 49 is a distraction, the focus here should be on the compound, its mechanism, and the limitations due to toxicity.

There are numerous cases of redundancy - e.g. the discussion of ABT-594 starting at line 84 and again at line 194. The section from lines 84 -104 should be integrated into section 3.

2 .Pharmacocokinetics - the discussion of epibatidine pharmacokinetics and metabolism can be made more concise - the take-away message both for epibatidine and flubatine is that there is very little metabolism. It would be helpful to include the structure of flubatine.

3. Given the side-effects of ABT-594, is there a future for therapeutic use of epibatidine analogs as analgesics? Or is the main use likely to be as tracers for diseases such as AD, where toxicities are not an issue due to the trace amounts used?

Author Response

Comments and Suggestions for Authors

This manuscript reviews the biology of epibatidine, a natural alkaloid, and its synthetic derivatives. Epibatidine has analgesic effects through α4β2 nicotinic acetylcholine receptors, and could have potential as a non-addictive alternative to opioid analgesic agents. However, its use as an analgesic is limited by toxicity mediated through CNS and skeletal neuro-muscular junction α7 nicotinic receptors, which can cause seizures and muscle paralysis. The resultant narrow therapeutic window could be widened by the development of derivatives with improved selectivity towards the α4β2 acetylcholine receptor.

The manuscript contains a considerable amount of information about epibatidine and its derivatives, but the value is reduced by a lack of organization and by redundancy. The impression given is that a number of authors have contributed sections but no-one has taken the time to synthesize the information adequately into a coherent document. In addition, given the toxicity of epibatidine, it would seem appropriate to focus more on the derivatives and less on the parent compound.

Answer: Thank you for your comments. We changed the organization of the manuscript focusing also on the toxicity but, first of all re-organizing the paragraphs, discussing more appropriately the analogs of epibatidine as therapeutic tools.

Abstract: The abstract should include the limitations of epibatidine due to toxicity and the potential of synthetic analogs as analgesic agents through improved selectivity. The metabolism of epibatidine is of minor interest and could be omitted from the abstract.

Answer: We followed all the suggestions of the Reviewer: we include the limitations of epibatidine due to toxicity, the metabolism has been omitted in the abstract. Thanks.

Introduction: The introduction lacks focus. e.g. the introduction of synthetic strategies in Line 49 is a distraction, the focus here should be on the compound, its mechanism, and the limitations due to toxicity.

Answer: Done 

There are numerous cases of redundancy - e.g. the discussion of ABT-594 starting at line 84 and again at line 194. The section from lines 84 -104 should be integrated into section 3.

Answer: Done

2 .Pharmacocokinetics - the discussion of epibatidine pharmacokinetics and metabolism can be made more concise - the take-away message both for epibatidine and flubatine is that there is very little metabolism. It would be helpful to include the structure of flubatine.

Answer: Thank you. We have expressed the results of some studies in a more concise way.

3. Given the side-effects of ABT-594, is there a future for therapeutic use of epibatidine analogs as analgesics? Or is the main use likely to be as tracers for diseases such as AD, where toxicities are not an issue due to the trace amounts used?

Answer: Discussion about therapeutic use of epibatidine analogues were added in the last section.

Reviewer 3 Report

Putting this in an open access journal might be helpful for new researcher. However, there are more info about epibatidine that could have been included and its comparison to other drugs.

Line 228-229: ... by one hand, a correlation between a reduced nAChRs expression in
... by the other hand, an increase in nAChR density   should read

Line 228-229: ...on one hand, a correlation between a reduced nAChRs expression in
... on the other hand, an increase in nAChR density

Author Response

Comments and Suggestions for Authors

Putting this in an open access journal might be helpful for new researcher. However, there are more info about epibatidine that could have been included and its comparison to other drugs.

Line 228-229: ... by one hand, a correlation between a reduced nAChRs expression in

...by the other hand, an increase in nAChR density   should read

Line 228-229: ...onone hand, a correlation between a reduced nAChRs expression in

...onthe other hand, an increase in nAChR density

Answer: Done, thanks.

Round  2

Reviewer 2 Report

No additional comments